# Early serum tumor marker levels after fourteen days of tyrosine kinase inhibitor targeted therapy predicts outcomes in patients with advanced lung adenocarcinoma

Hung-Jen Chen [1,2,3]*, Chih-Yen Tu[1,2,3], Kuo-Yang Huang[4], Chun-Ru Chien[2,3,5], Te-Chun Hsia[1,2]

1 Division of Pulmonary and Critical Care Medicine, Taichung, Taiwan, 2 China Medical University Hospital, Taichung, Taiwan, 3 School of Medicine, College of Medicine, China Medical University, Taichung, Taiwan, 4 Division of Chest Medicine, Department of Internal Medicine, Changhua Christian Hospital, Changhua, Taiwan, 5 Department of Radiation Oncology, China Medical University Hospital, Taichung, Taiwan

* redman0127@gmail.com

## Abstract

### Objective

Image evaluation strategy for lung cancer patients has difficulty obtaining the appropriate quantity of diffuse lung nodules and bone metastases. The study was to demonstrate whether early variations in the levels of serum 4-tumor markers (4-TMs)(carcinoembryonic antigen [CEA], cancer antigen [CA]125, CA19-9, and CA15-3) after TKI targeted therapy were associated with treatment response in patients with lung adenocarcinoma.

### Methods

Patients with stage IIIB-IV lung adenocarcinoma taking epidermal growth factor receptor (EGFR) TKIs or anaplastic lymphoma kinase (ALK) inhibitors were enrolled prospectively from June 2012 to February 2015. According to the variations of the percentage of change in 4-TM levels (4-TMpc), we divided patients into ascending (increases in 4-TMpc over the 7th- 14th day) and descending (decreases in 4-TMpc over the 7th- 14th day) groups.

### Results

184 patients were enrolled, and 89% had at least one of the pre-treatment evaluable TMs and were further analyzed. An excellent response to the TKI targeted therapy was accurately predicted in the descending group, as determined using receiver operating characteristic curve analysis (an area under the curve, 0.83). Multivariate Cox hazards model analyses demonstrated that the type of 4-TMpc and mutation status were the strongest predictors of progression-free survival (PFS)(descending versus ascending, hazard ratios [HR] 0.30, 95% confidence interval [CI], 0.19–0.47; sensitive mutation versus wide type, HR 0.30, 95% CI, 0.19–0.48).

**Data Availability Statement:** All relevant data are within the paper and its Supporting Information files.

**Funding:** The authors received no specific funding for this work.

**Competing interests:** The authors have declared that no competing interests exist.

**Abbreviations:** EGFR, Epidermal growth factor receptor; ALK, anaplastic lymphoma kinase; TKI, tyrosine kinase inhibitor; PFS, progression-free survival; RECIST, Response Evaluation Criteria in Solid Tumors; TM, tumor marker; CEA, carcinoembryonic antigen; NSCLC, non-small cell lung cancer; CA125, carbohydrate antigen125; CA19-9, carbohydrate antigen 19–9; CA15-3, carbohydrate antigen153; PCR, polymerase chain reaction; IHC, Immunohistochemistry; $TM_v$, tumor markers with inherent intra-individual biological variation and within-laboratory coefficients of variation; 4-$TM_{pc}$, percent of change of 4 tumor markers; $TM_p$, previous TM; $TM_l$, later TM; CXR, chest radiograph; CT, computed tomography; PR, partial response; SD, stable disease; $SD_{-30}$, stable disease with tumor reduction less than 30%; $SD_{+20}$, stable disease with tumor increasing less than 20%; PD, progressive disease; ROC, receiver operating characteristic; AUC, area under the curve; CI, confidence interval; HR, hazard ratio.

## Conclusions

Type of 4-TMpc 14 days after TKI targeted therapy is associated with an image response and PFS, without regarding mutation status, in patients with advanced lung adenocarcinoma.

## Introduction

The prognosis of advanced lung adenocarcinoma patients with genotype-driven mutations has improved due to targeted therapy [1–5]. Epidermal growth factor receptor (*EGFR*) mutation and anaplastic lymphoma kinase (*ALK*) rearrangement are two major oncogenic alterations that are targeted with available tyrosine kinase inhibitors (TKIs). *EGFR* TKIs, gefitinib [1], erlotinib [2], and afatinib [3], and ALK inhibitors, crizotinib [4] and ceritinib [5], have prolonged progression-free survival (PFS) rates in advanced lung adenocarcinoma patients with sensitive *EGFR* mutations and *ALK* rearrangement, respectively.

However, although sensitive *EGFR* mutations and *ALK* rearrangement are strong predictors of good response to TKIs targeted therapy, not all patients (about 60–70%) respond to the therapy [1–5], although a portion of patients with *EGFR* wild type mutations and *ALK*-negative have shown a response [6,7].

Morphologic imaging studies using the Response Evaluation Criteria in Solid Tumors (RECIST) remain the standard tool for evaluating treatment response [8]. However, this image evaluation strategy has several limitations, such as difficulty in obtaining the appropriate quantity of diffuse lung nodules, pleural effusions, and bone metastases [9].

Serum tumor marker (TM) concentration is a reflection of the synthesis potential of the tumor [10]. Assessment of TMs for evaluating treatment responses is clinically objective. Elevated carcinoembryonic antigen (CEA) levels have been observed in 40–80% of patients with non-small cell lung cancer (NSCLC) [11,12]. Nevertheless, a single assessment of CEA levels to evaluate lung cancer treatment response is not sensitive [9].

Thus far, not much is known about TM levels' changes and their genuine relationship to predict prognosis in adenocarcinoma patients receiving TKI targeted therapy [9,13,14]. We assessed 4-TMs, CEA, carbohydrate antigen (CA) 125 (CA125), CA19-9, and CA15-3. The selection of these tumor markers was based on reports [9–14] and our previous pilot study results (data not shown). There 4-TMs were also easy to assess in clinical practice and provided the most cost-effective coverage in patients with advanced lung adenocarcinoma. The study aimed to demonstrate whether early variations in serum 4-TMs after TKI targeted therapy were associated with treatment response and PFS in advanced lung adenocarcinoma patients.

## Materials and methods

### Patient selection and treatment

Patients with stage IIIB-IV lung adenosquamous cancer or adenocarcinoma taking *EGFR* TKIs (gefitinib, erlotinib, or afatinib) or *ALK* inhibitors (crizotinib or ceritinib) in different lines of therapy were enrolled in a prospective, single-center at the China Medical University Hospital from June 2012 to February 2015. We calculated the sample size needed for the kappa analysis by PASS software (version 20.0.1, NCSS, LLC. Kaysville, Utah, USA) via assuming that the proportion of good response to being around 50% in patients with advanced lung adenocarcinoma treated with targeted therapy. A sample of 161 patients achieves 90% power to

detect a true Kappa value of 0.60 in a test of H0: Kappa = 0.40 vs. H1: Kappa>0.40, at a significance level of 0.05. Furthermore, considering that 10–15% of patients cannot be adapted to TM due to biochemical non-accessibility, we increased the sample size to around 190 patients. The study was approved by the China Medical University Hospital Institutional Review Board, Taichung, Taiwan (CMUH DMR 101-IRB1-087 and CMUH 104-REC1-108). Written informed consent was obtained from all patients.

### *EGFR* mutation and *ALK* immunohistochemistry analysis

The tumor DNA sequences of exons 18 to 21 of *EGFR* were determined using direct forward and reverse sequencing via the polymerase chain reaction (PCR) product from nested PCR reactions [15]. Sensitizing mutations are defined as G719X in exon 18, in-frame deletions or insertion of exon 19, A763_Y764 insFQEA mutation, and S768I in exon 20 and L858R or L861Q in exon 21 [16–18].

ALK immunohistochemistry (IHC) was performed using the Ventana anti-ALK (D5F3) CDx assay. The staining results were evaluated using a binary scoring system: positive or negative following the manufacturer's instructions [19].

### Serum CEA, CA125, CA19-9, and CA15-3 level detection and analysis

TM levels obtained from peripheral blood samples were measured before TKI targeted therapy and after 7 and 14 days of treatment. To reduce TM levels' influence with inherent intra-individual biological variation and within-laboratory coefficients of variation ($TM_v$) [20–22], we defined a cutoff level for each individual using pre-treatment TM levels of 2-fold over the standard upper limit. Therefore, enrollment criteria included CEA, CA125, CA19-9, and CA15-3 levels at 10.0 ng/mL, 70 units/mL, 70 units/mL, and 76 units/mL, respectively. Patients who did not show an elevation in TMs above this level were regarded as biochemically non-assessable and were excluded from further follow-up.

To evaluate changes in TM levels after TKIs targeted therapy and to account for patients having more than one evaluable TM, we created a formula "percentage of change of 4 TMs (4-$TM_{pc}$)". Assuming a distinct sub-clone released each TM within the tumor bulk (1-marker (later)/marker(previous)), a reasonable estimate of the proportion of tumor treatment for this sub-clone was made. Our Eq 1 represents the weighted average of the proportion of tumor treatment across different sub-clones.

$$4-TM_{pc} = \frac{\text{Total numbers of evaluable TMs} - \left(\frac{CEA_l^*}{CEA_p} + \frac{CA125_l^*}{CA125_p} + \frac{CA199_l^*}{CA199_p} + \frac{CA153_l^*}{CA153_p}\right)}{\text{Total numbers of evaluable TMs}} = \times 100 \qquad Eq-1$$

[note: subscript "p" = previous; subscript "l" = later; *2-fold over the standard upper limit was regarded as evaluable.]

For example, if on Day 0, a patient had the following serum TM values (CEA 2 ng/mL, CA125 225 units/mL, CA19-9 5 units/mL, and CA15-3 197 units/mL), then this patient had only 2 evaluable TMs (CA125 and CA15-3). If on Day 7, the TM levels were CA125 175 units/mL and CA15-3 132 units/mL, then the 4-$TM_{pc}$ over the $0^{th}$-$7^{th}$ day was 27.6%, as shown in Eq 2.

$$27.6\% = \frac{2 - \left(\frac{CA125(175)}{CA125(225)} + \frac{CA153(132)}{CA153(197)}\right)}{2} \times 100\% \qquad Eq-2$$

According to variations in 4-$TM_{pc}$ on days 0, 7, and 14, we divided patients into four

groups. Type 1. Ascending: patients who sustained an increase in 4-TM$_{pc}$. Type 2. Descending-ascending: patients who showed a decreasing trend on the 7$^{th}$ day, and then showed an increasing trend on the 14$^{th}$ day. Type 3. Ascending-descending: patients who showed an increasing trend on the 7$^{th}$ day and then showed a decreasing trend at subsequent time points. Type 4. Descending: patients who showed a persistently decreased 4-TM$_{pc}$. For minimizing TM$_v$ interference, the following definitions were created [20–22]: when 4-TM$_{pc}$ was <5% over the 7$^{th}$-14$^{th}$ day, and it was defined as "type uncertain." Confirmed decreases in 4-TM$_{pc}$ over the 7$^{th}$- 14$^{th}$ day (types 3/4) were consistent with tumor response. Similarly, increases in 4-TM$_{pc}$ over the 7$^{th}$-14$^{th}$ day (types 1/2) were regarded as tumor progression.

## Imaging-based response

Tumor response was assessed on chest radiographs (CXR) and computed tomography (CT) scans, using the RECIST version 1.1 in an independent radiologic review by assessors who did not know the results of 4-TM$_{pc}$ studies and confirmed at least two scans obtained 28 days apart. Long-term follow-up was performed until July 31, 2015.

Therapeutic efficacy was classified as partial response (PR), stable disease (SD) with tumor reduction <30% (SD$_{-30}$), SD with the increase in tumor size <20% (SD$_{+20}$), or progressive disease (PD). Patients who died due to cancer between these CT/CXR procedures were classified as having PD. While analyzing the correlation between 4-TM$_{pc}$ after 14 days TKI targeted therapy and RECIST assessed response, and we combined categories of PR and SD$_{-30}$ into the "good response group." In contrast, the "poor response group" included cases with PD and SD$_{+20}$. This classification's rationales were that: (1) It was the straight forward approach to classify responders vs. non-responders. (2) It was also difficult for PR, SD, or PD to correlate with the ascending or descending of the 4-TMpc from a statistical point of view. The endpoint was PFS. PFS was assessed from the date of the beginning of TKI targeted therapy to the date of PD or death due to cancer. If a patient was lost to follow-up or had no event, time to progression was censored as the date of the last contact date.

## Statistical analysis

To obtain a descriptive analysis, we resumed each continuous variable as median and 25–75, and categorical variables as proportion. We performed receiver operating characteristic (ROC) curve analysis for 4-TMs to predict TKI targeted therapy's response. The agreement between the 4-TMs and the image-based morphologic response was evaluated using the kappa statistic. PFS was analyzed according to the Kaplan-Meier method and was compared with the log-rank test. Cox proportional hazards model was used to evaluate independent predictive factors associated with PFS. Data were analyzed using SPSS-17 (IBM SPSS Statistics. Inc. Chicago, IL, USA). For all analyses, two-sided P<0.05 was taken as statistically significant.

## Results

In all, 195 patients with a diagnosis of stage IIIb-IV adenocarcinoma (one with adenosquamous carcinoma) were screened for 4-TM levels before the start of TKI targeted therapy, and 11 were excluded: 1 because the patient had severe interstitial lung disease after taking erlotinib and ten because the standard protocol was not followed. Seven patients were recruited more than once because patients accepted re-challenge TKI targeted therapy. Therefore, 184 patients and 191 patient-times were enrolled in this study, including 29 accepted the diagnostic procedure of computed tomography-guided core needle biopsy, 68 accepted transbronchial biopsy, 20 accepted ultrasound-guided biopsy, 27 accepted thoracentesis or pericardiocentesis, and 40 accepted operative procedures. Detailed baseline characteristics are summarized in Table 1.

**Table 1. Baseline characteristics of patients.**

| Variables | All patients | CEA/CA125/CA153/CA199 Elevation | CEA/CA125/CA153/CA199 No elevation |
|---|---|---|---|
| All patient times* | 191 | 170 | 21 |
| All patients | 184 | 163 | 21 |
| Age, yr | 62.5 (55.0–73.0) | 62.0 (55.0–72.0) | 68.0 (54.5–78.5) |
| Sex | | | |
| Male | 76 (41.3) | 68 (41.7) | 8 (38.1) |
| Female | 108 (58.7) | 95 (58.3) | 13 (61.9) |
| Smoking | | | |
| Never | 131 (71.2) | 118 (72.4) | 13 (61.9) |
| Former/current | 53 (28.8) | 45 (27.6) | 8 (38.1) |
| Stage | | | |
| IIIb | 5 (2.7) | 5 (3.1) | 0 |
| Iva | 65 (35.3) | 54 (33.1) | 11 (52.4) |
| IVb | 114 (62.0) | 104 (63.8) | 10 (47.6) |
| Performance status* | | | |
| 0–1 | 108 (56.6) | 95 (55.8) | 13 (61.9) |
| 2 | 26 (13.6) | 21 (12.4) | 5 (23.8) |
| 3–4 | 57 (29.8) | 54 (31.8) | 3 (14.3) |
| Mutation | | | |
| EGFR sensitizing mutation | | | |
| Exon 19 deletion | 65 (35.3) | 54 (33.1) | 11 (52.4) |
| L858R | 55 (29.9) | 47 (28.8) | 8 (38.1) |
| Others+ | 10 (5.4) | 10 (6.1) | 0 |
| ALK rearrangement | 5 (2.7) | 5 (3.1) | 0 |
| EGFR and ALK-negative | 44 (23.9) | 42 (25.8) | 2 (9.5) |
| Unknown | 5 (2.7) | 5 (3.1) | 0 |
| Targeted therapy* | | | |
| Gefitinib | 80 (41.9) | 70 (41.2) | 10 (77.7) |
| Erlotinib | 88 (46.1) | 78 (45.9) | 10 (47.6) |
| Afatinib | 19 (9.9) | 18 (10.6) | 1 (4.8) |
| Crizotinib | 2 (1.0) | 2 (1.2) | 0 |
| Ceritinib | 2 (1.0) | 2 (1.2) | 0 |
| Treatment-line* | | | |
| First-line | 167 (87.4) | 150 (88.2) | 17 (81.0) |
| Second-line | 23 (12.0) | 19 (11.2) | 4 (19.0) |
| Third-line | 1 (0.5) | 1 (0.6) | 0 |

*All patient times

+One patient had L858R + T790M. One patient had exon 19 deletion + T790M. One patient had exon 19 deletion + L858R. One patient had exon 19 deletion + S768I. One patient had S768I. Three patients had G719X. Two patients had L861Q.

EGFR epidermal growth factor receptor, ALK anaplastic lymphoma kinase

CEA carcinoembryonic antigen, CA125 carbohydrate antigen 125, CA19-9 carbohydrate antigen 19–9, CA15-3 carbohydrate antigen 15–3

Activating mutations were documented in 73% cases, and 3% had an unknown mutation status because no sufficient pathological material was available. Further, 98% of patient-times received *EGFR* TKIs, and 2% received *ALK*-inhibitors

Of the 184 enrolled patients, baseline serum CEA levels, CA125, CA19-9, and CA15-3 were 2-fold over the standard upper limit in 71%, 62%, 27%, and 23% patients, respectively. In all, 89% of patients with at least one of the pre-treatment evaluable TM were further analyzed. Of

these, 7% had all 4 TM levels elevated, 26% had 3, 33% had 2, and 34% had 1 TM elevated. The summarized data is shown in Table 2.

Of 170 patient-times after TKI targeted therapy with evaluable TMs, PR, $SD_{-30}$, $SD_{+20}$, and PD were 55%, 23%, 1% and 21%, respectively. We divided the patients into 4 groups by 14 days 4-$TM_{pc}$: type 1, 22%; type 2, 4%; type 3, 25%; and type 4, 42%. Further, 8% of patients were classified as "type uncertain." Types 1/2 were observed in 32 of 44 patients (73%) who showed a poor response. On the contrary, types 3/4 were observed in 111 of 113 patients (98%), who showed a good response (Fig 1). The presence of types 3/4 could accurately predict "good response" by using ROC curve analysis, with an area under the curve (AUC) 0.83 (95% confidence interval [CI] 0.73 to 0.93). The Kappa value between 157 cases with the type of 4-$TM_{pc}$ and measurable radiographic lesions was 0.762 (P<0.001) (Fig 1).

However, 24% of patients had baseline CEA< 10 ng/mL. This group of patients could not be biochemically assessed using CEA levels alone. The AUC was only 0.51 (95% CI, 038–0.63) for predicting TKI targeted therapy response with the type of CEA. The Kappa value in 117 cases was 0.449 (Fig 2).

One hundred and forty patients in whom first-time TKI targeted therapy with the type of 4-$TM_{pc}$ was used, were further analyzed using Kaplan-Meier curves for PFS. As shown in Fig 3A, the median PFS had no significant difference between types 1 and 2 (30 and 28 days), nearly the same as types 3 and 4 (252 and 245 days). However, PFS was significantly longer in

**Table 2. a. Characteristics of carcinoembryonic antigen (CEA), cancer antigen (CA) 125, CA19-9, and CA15-3 in 184 patients with advanced lung adenocarcinoma. b. Items of 163 patients with advanced lung adenocarcinoma with evaluable tumor markers.**

| a | | |
|---|---|---|
| Variables | N (%) | Median (25–75) |
| CEA ng/mL | 130 (70.7) | 121.2 (32.1–345.6) |
| CA125 units/mL | 114 (62.0) | 167.1 (112.8–360.2) |
| CA19-9 units/mL | 50 (27.2) | 334.6 (159.5–1423.5) |
| CA15-3 units/mL | 42 (22.8) | 155.3 (90.6–253.2) |
| b | | |
| Variables | | N (%) |
| One tumor marker | CEA | 38 (23.3) |
| | CA125 | 17 (10.4) |
| | CA19-9 | 0 |
| | CA15-3 | 1 (0.6) |
| Two tumor markers | CEA + CA125 | 32 (19.6) |
| | CEA + CA15-3 | 5 (3.1) |
| | CEA + CA19-9 | 3 (1.8) |
| | CA125 + CA15-3 | 5 (3.1) |
| | CA125 + CA19-9 | 7 (4.3) |
| | CA15-3 + CA19-9 | 1 (0.6) |
| Three tumor markers | CEA + CA125 + CA15-3 | 15 (9.2) |
| | CEA + CA125 + CA19-9 | 24 (14.7) |
| | CEA + CA15-3 + CA19-9 | 1 (0.6) |
| | CA125 + CA15-3 + CA19-9 | 2 (1.2) |
| Four tumor markers | CEA + CA125 + CA15-3 + CA19-9 | 12 (7.4) |

CEA carcinoembryonic antigen, CA125 carbohydrate antigen 125, CA19-9 carbohydrate antigen 19–9, CA15-3 carbohydrate antigen 15–3

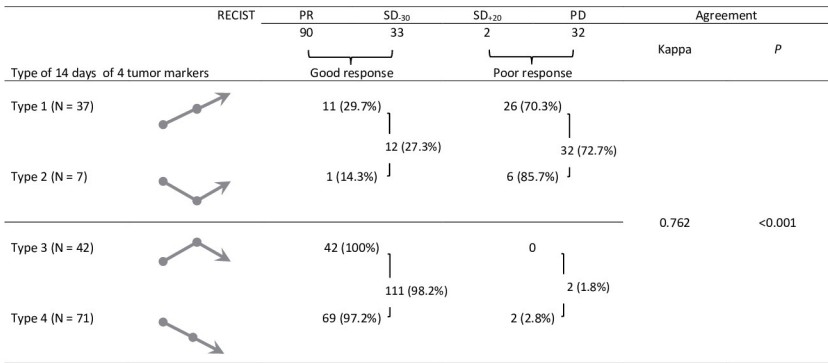

**Fig 1. Relevance between image-based response and the type of 4 tumor marker levels in 157 patients with advanced lung adenocarcinoma.** RECIST Response Evaluation Criteria in Solid Tumors, PR partial response, $SD_{-30}$ stable disease with tumor reduction <30%, $SD_{+20}$ stable disease with tumor increasing <20%, PD progressive disease.

types 3/4 than in types 1/2 (*P*<0.001). Analysis for subgroups stratified according to mutation status found that PFS was still longer in patients with types 3/4 than in patients with types 1/2 (activated mutation group, *P* = 0.016 (Fig 3B); *EGFR* and *ALK*-negative/unknown group, *P*<0.001 (Fig 3C)).

In the univariate analysis using the Cox hazards model, types 3/4 and sensitive mutation were the only two PFS predictive factors (*P*<0.001). Multivariate Cox hazard model analyses shows the same result (types 3/4 versus types 1/2, *P*<0.001, hazard ratio (HR) 0.30, 95% CI, 0.19–0.47; sensitive mutation versus *EGFR* and *ALK*-negative/unknown, *P*< 0.001, HR 0.30, 95% CI, 0.19–0.48) (Table 3).

## Discussion

This prospective study was the first to provide CEA, CA125, CA19-9, and CA15-3 permutation and combination in patients with advanced lung adenocarcinoma. We compared the AUC of CEA and 4-TMs to predict clinical responses of TKI targeted therapy. Our study demonstrated that 14 days of 4-TM$_{pc}$ type could be an early predictor of TKI targeted therapy efficacy. The descending type of 4-TM$_{pc}$ over the $7^{th}$- $14^{th}$ days had longer PFS in mutated or non-mutated adenocarcinoma patients.

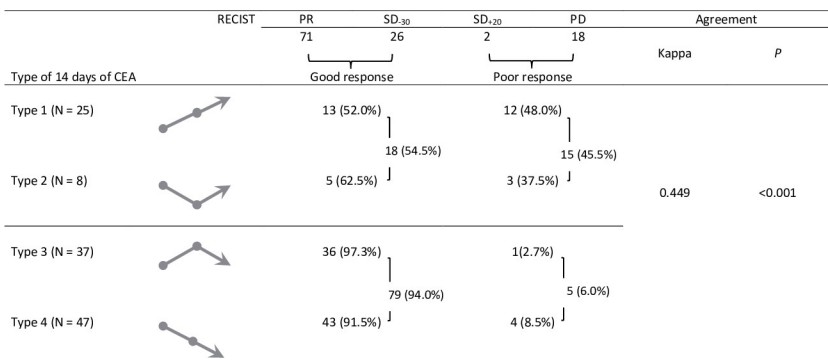

**Fig 2. Relevance between image-based response and type of CEA percentage of change over the $0^{th}$-$14^{th}$ day in 117 patients with advanced lung adenocarcinoma.** CEA carcinoembryonic antigen, RECIST Response Evaluation Criteria in Solid Tumors, PR partial response, $SD_{-30}$ stable disease with tumor reduction <30%, $SD_{+20}$ stable disease with tumor increasing <20%, PD progressive disease.

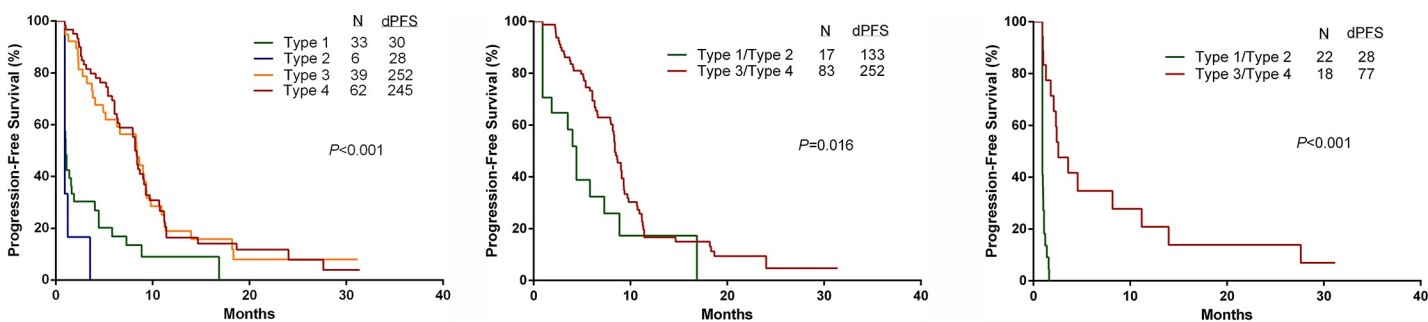

**Fig 3.** Kaplan-Meier curves for progression-free survival (**A**) in the entire cohort, (**B**) in activated mutation, (**C**) in *EGFR* and *ALK*-negative/unknown groups, respectively. EGFR epidermal growth factor receptor, ALK anaplastic lymphoma kinase, d-PFS progression-free days.

Although squamous cell lung cancer and adenocarcinoma are a subset of NSCLC, they have different driver mutations and treatment [23]. CEA is more frequently reported in patients with adenocarcinoma than squamous lung cancer [11,12]. Therefore, we focused on lung adenocarcinoma. Furthermore, not all lung adenocarcinoma patients had elevated CEA levels [9,13,14]. Advanced lung adenocarcinoma has other potentially valuable TMs in addition to CEA. In our series, CA125, CA 19–9, and CA15-3 levels reached evaluable criteria in 62%, 27%, and 23% patients, respectively. While combined with 4-TMs, only 11% of patients had 4-TMs below the evaluable levels (Table 2).

There is insufficient evidence to support a conclusion concerning the standardized combination of TMs to evaluate tumor status. Different intra-tumor sub-clones may release different TMs. We presumed "one TM, one evaluable clone" and combined TMs as "4-TM$_{pc}$".

Zhang et al. indicated that the descending type of CEA within one month correlated with PR and SD of *EGFR*-TKI in patients with lung adenocarcinoma [13]. However, the CEA type can be affected by TM$_v$ and can influence the clinician to make a wrong decision [20,22]. In our 170 patient-times with evaluable TM cohort, "type uncertain" was classified when 4-TM$_{pc}$

**Table 3. Univariate and multivariate prediction of progression-free survival.**

| | Univariate | | | Multivariate | | |
|---|---|---|---|---|---|---|
| | **HR** | **95% CI** | ***p* value** | **HR** | **95% CI** | ***p* value** |
| Age (years) | | | | | | |
| <65 vs. ≥65 | 1.25 | 0.87–1.80 | 0.24 | 1.56 | 1.05–2.34 | .030 |
| Gender | | | | | | |
| Female vs. Male | 0.85 | 0.59–1.23 | 0.39 | 1.17 | 0.73–1.88 | .516 |
| Smoking habit | | | | | | |
| Never vs. Current/former | 0.70 | 0.47–1.06 | 0.09 | 0.82 | 0.48–1.40 | .469 |
| Performance status | | | | | | |
| 0–1 vs. ≥2 | 0.89 | 0.61–1.30 | 0.55 | 1.03 | 0.70–1.52 | .881 |
| Stage | | | | | | |
| IIIb/IVa vs. IVb | 0.79 | 0.54–1.16 | 0.22 | 0.89 | 0.60–1.33 | .572 |
| Mutation | | | | | | |
| MUT vs. WT/UNK | 0.34 | 0.23–0.52 | <0.001 | 0.30 | 0.19–0.48 | <0.001 |
| CEA/CA125/19-9/15-3 | | | | | | |
| Type 3/4 vs. Type 1/2 | 0.29 | 0.19–0.44 | <0.001 | 0.30 | 0.19–0.47 | <0.001 |

CI confidence interval, HR hazard ratio, MUT mutated patients, WT/UNK wild-type/unknown patients, CEA carcinoembryonic antigen, CA125 carbohydrate antigen 125, CA19-9 carbohydrate antigen 19–9, CA15-3 carbohydrate antigen 15–3

over the 7th- 14th day was <5%. We divided the others into four groups. Among treatment, patients showing effectiveness may have an ascending 4-TM$_{pc}$ pattern before the descending pattern (type 3). This transient increase in TM levels is known as surges [13,24]. Otherwise, type 2 with fluctuation in 4-TM$_{pc}$ and then an ascending pattern the over 7th-14th day is not logical if TKI targeted therapy is considered effective (Fig 1).

We evaluated the type of 4-TM$_{pc}$ within 14 days of TKI targeted therapy and the relationship with imaging-based response and PFS. The ROC curve analysis showed that using 4-TMs for predicting the efficacy of TKI targeted therapy response had a higher AUC (0.83) than that using CEA (0.51). The Kappa value for the agreement analysis between 157 cases with the type of 4-TM$_{pc}$ and radiographic results was "good" (0.762) (Fig 1). However, using CEA levels, 24% of patients were biochemically non-assessable in this 170 patient-time series. The kappa value was 0.449 only (Fig 2). These findings showed that using 4-TMs to predict TKI targeted therapy response was more accurate than using CEA alone.

The results of sensitizing mutations cannot guarantee a clinical response to TKI targeted therapy [6,7,25]. It is essential to develop a new strategy for early prediction of the effect of TKI targeted therapy. In our series, patients with types 3/4, 4-TM$_{pc}$ had a longer PFS than those with types 1/2. Regarding the ascending (type 3) or descending pattern (type 4) over the 0th-7th day, patients' outcomes were similar. On the contrary, similar PFS was observed in patients with types 1 and 2 (Fig 3A). Regarding activated mutation (Fig 3B) and *EGFR* and *ALK*-negative/unknown group (Fig 3C), PFS was also significantly longer in patients with types 3/4 than in patients with types 1/2. Similarly, in multivariate models, our results demonstrate that 4-TM$_{pc}$ and mutation status continues to be the strongest predictors of PFS (Table 3).

Our study's strengths include its prospective design, and a large number of patients included compared to previous TM studies in NSCLC patients under TKI targeted therapy [9,13,14]. However, certain drawbacks should be considered. Firstly, it is a single-center study. Secondly, there is insufficient evidence to define the best cutoff level for minimizing the interference of TM$_v$. Thirdly, we did not extend our study after Feb 2015 because the enrolled cases and the follow-up time are sufficient to reflect the study results in which we did find high-level agreement [kappa = 0.762] by planned accrual [around 190 cases]. Furthermore, the investigated scenario (gefitinib, erlotinib, afatinib, crizotinib or ceritinib for lung adenocarcinoma patients) was still valid in Taiwan in 2020, although whether our finding was applicable for some new inhibitors (such as osimertinib [26] or alectinib [27]) deserved to be studied.

## Conclusion

In conclusion, image evaluation strategy with RECIST for patients with lung cancer has difficulty in obtaining the appropriate quantity of diffuse lung nodules, pleural effusions, and bone metastases. The type of 4-TM$_{pc}$ after 14 days TKI targeted therapy is associated with image response and PFS without accounting for mutation status in advanced lung adenocarcinoma patients. Our study results could help early therapeutic decision making by identifying patients who may benefit from gefitinib, erlotinib, afatinib, crizotinib, or ceritinib 14 days after TKI targeted therapy.

## Supporting information

**S1 Data.**
(XLS)

## Acknowledgments

The authors wish to thank Chia-Ing Li for her technical help with statistical analysis.

## Author Contributions

**Conceptualization:** Hung-Jen Chen.

**Data curation:** Hung-Jen Chen, Kuo-Yang Huang.

**Formal analysis:** Hung-Jen Chen.

**Methodology:** Hung-Jen Chen.

**Resources:** Chih-Yen Tu, Te-Chun Hsia.

**Writing – original draft:** Hung-Jen Chen, Chun-Ru Chien.

**Writing – review & editing:** Hung-Jen Chen.

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
