## [Decision Letter · Decision Letter 0]

13 Aug 2020

PONE-D-20-12186

Early Serum Tumor Marker Levels After Fourteen Days of Tyrosine Kinase Inhibitor Targeted Therapy Predicts Outcomes in Patients with Advanced Lung Adenocarcinoma

PLOS ONE

Dear Dr. Chen

Thank you for submitting your manuscript to PLOS ONE. After careful consideration, we feel that it has merit but does not fully meet PLOS ONE’s publication criteria as it currently stands. Therefore, we invite you to submit a revised version of the manuscript that addresses the points raised during the review process.

Please address the reviewer comments as outlined in attached information.

We look forward to receiving your revised manuscript.

Kind regards,

Christina L Addison, Ph.D.

Academic Editor

PLOS ONE

Journal Requirements:

2. Please provide a sample size and power calculation in the Methods, or discuss the reasons for not performing one before study initiation.

3. In the Methods section, please provide the source of the anti-ALK antibody used for your study.

4. Thank you for inncluding your funding statement;"The funders had no role in study design, data collection and analysis, decision to publish, or preparation of the manuscript."

5. 

In your Data Availability statement, you have not specified where the minimal data set underlying the results described in your manuscript can be found. PLOS defines a study's minimal data set as the underlying data used to reach the conclusions drawn in the manuscript and any additional data required to replicate the reported study findings in their entirety. All PLOS journals require that the minimal data set be made fully available. For more information about our data policy, please see http://journals.plos.org/plosone/s/data-availability.

6. Please include your tables as part of your main manuscript and remove the individual files. Please note that supplementary tables (should remain/ be uploaded) as separate "supporting information" files

Reviewers' comments:

Reviewer's Responses to Questions

**Comments to the Author**

1. Is the manuscript technically sound, and do the data support the conclusions?

Reviewer #1: Yes

Reviewer #2: Yes

2. Has the statistical analysis been performed appropriately and rigorously? 

Reviewer #1: Yes

Reviewer #2: Yes

3. Have the authors made all data underlying the findings in their manuscript fully available?

Reviewer #1: Yes

Reviewer #2: Yes

4. Is the manuscript presented in an intelligible fashion and written in standard English?

Reviewer #1: Yes

Reviewer #2: Yes

5. Review Comments to the Author

Reviewer #1: Not all lung adenocarcinomas with sensitive mutations can benefit from targeted therapy. How to distinguish effective and ineffective patients early in patients with targeted therapy has certain clinical significance. The author of the manuscript tried to answer this question through analyzing changes of tumor markers, which gave us some inspirations. There are three questions as follows. Firstly, in manuscript, the authors evaluated the efficacy of targeted therapy by imaging-based response according to the RECIST standard. Why are the patients whose efficacy evaluation are SD divided into SD-20 and SD+30, instead of distinguishing by PR, SD, PD or DCR? It needs to be explained. Secondly, the references at the end of the manuscript seem to be inconsistent with the label in the article. For example, the reference 11 in the article seems to be 13 at the end, and it needs to be revised. Thirdly, cases in this study are all from 2012 to 2015, why are there no recent patients enrolled to increase the sample size for further statistical analysis?

Reviewer #2: It is difficult to evaluate efficacy of treatment in patients with lung cancer in some cases. The present study　investigated the association between TKI treatment and tumor markers in advanced lung cancer. Overall, the study is well performed, however, there are potential limitations in this study. First, there was no mention of the reason why the authors chose those tumor markers. Second, a method of obtaining pathological samples in patients with lung cancer was not described in this study. Third, some patients received TKIs regardless of unknown mutation status. Is it ethically acceptable? I would like to recommend the minor revisions for this manuscript.

6. PLOS authors have the option to publish the peer review history of their article (what does this mean?). If published, this will include your full peer review and any attached files.

Reviewer #1: **Yes: **Jinliang Wang

Reviewer #2: No

---

## [Author Response · Author response to Decision Letter 0]

2 Sep 2020

Revision of PONE-D-20-12186.R1

Dear editor, 

 Thank you for the nice and detailed review. We have made point - by - point revisions and responses to all reviewers’ comments. The reasons and revisions are listed in the following tables to each of the comments, respectively. In the revised manuscript, all the changes are highlighted in RED color. We deeply appreciate your valuable comments, which stimulated a more thorough consideration of the paper. Thank you. 

Sincerely,

Hung-Jen Chen, MD

Division of Pulmonary and Critical Medicine, Department of Internal Medicine,

China Medical University Hospital, Taichung, Taiwan

No. 2, Yude Road, North District, Taichung City 40447, Taiwan 

Phone No.: +886-4-22052121#4665

Email: redman0127@gmail.com

 

Comments to the Author

Editor Q1.

# Editor Q1. Please ensure that your manuscript meets PLOS ONE's style requirements, including those for file naming. The PLOS ONE style templates can be found at

Reply

# In response to Editor Q1: We thanked the editor to remind the format issue. We had revised the manuscript via using the PLOS ONE style templates. We also highlighted other major changes in RED so the editor and reviewers can easily see where we made significant revision.

Editor Q2.

# Editor Q2. Please provide a sample size and power calculation in the Methods, or discuss the reasons for not performing one before study initiation.

Reply

# In response to Editor Q2: We apologized for incomplete description of sample size issue in the previous manuscript. We used the kappa analysis to evaluate the agreement of two methods (image-based response and tumor marker (TM) based) to determined the sample size. The sample size needed for the kappa analysis was calculated by PASS software (version 20.0.1, NCSS, LLC. Kaysville, Utah, USA). Assuming that the proportion of good response is around 50% in patients with advanced lung adenocarcinoma with targeted therapy, a sample of 161 patients achieves 90% power to detect a true Kappa value of 0.60 in a test of H0: Kappa=0.40 vs. H1: Kappa>0.40, at a significance level of 0.05. The kappa value is often interpreted as 0.21 to 0.40, indicating fair, and 0.41 to 0.60 moderate level of agreement. Furthermore, considering that 10-15% of patients cannot be adapted to TM due to biochemical non-accessibility, we increased the sample size to 190 patients. These were clarified in our revised manuscript [method P7. Line 106 - 114] 

Change in the text (See Page 7, Line 106-114) 

We calculated the sample size needed for the kappa analysis by PASS software (version 20.0.1, NCSS, LLC. Kaysville, Utah, USA) via assuming that the proportion of good response to being around 50% in patients with advanced lung adenocarcinoma treated with targeted therapy. A sample of 161 patients achieves 90% power to detect a true Kappa value of 0.60 in a test of H0: Kappa=0.40 vs. H1: Kappa>0.40, at a significance level of 0.05. Furthermore, considering that 10-15% of patients cannot be adapted to TM due to biochemical non-accessibility, we increased the sample size to around 190 patients.

Editor Q3.

# Editor Q3. In the Methods section, please provide the source of the anti-ALK antibody used for your study.

Reply

# In response to editor Q3: I am sorry for the apparent mistake and thank the reviewer for the careful review. I had corrected it in the revised manuscript as “ALK immunohistochemistry (IHC) was performed using the Ventana anti-ALK (D5F3) CDx assay. The staining results were evaluated using a binary scoring system: positive or negative following the manufacturer’s instructions.” [see method P7. Line 124 - 126]

Change in the text (See Page 7, Line 124-126) 

ALK immunohistochemistry (IHC) was performed using the Ventana anti-ALK (D5F3) CDx assay. The staining results were evaluated using a binary scoring system: positive or negative following the manufacturer’s instructions.

Editor Q4.

# Editor Q4. Thank you for including your funding statement; “The funders had no role in study design, data collection and analysis, decision to publish, or preparation of the manuscript."

a. Please clarify the sources of funding (financial or material support) for your study. List the grants or organizations that supported your study, including funding received from your institution.

d. If you did not receive any funding for this study, please state: “The authors received no specific funding for this work.”

Reply

# In response to editor Q4: The authors received no specific funding for this work. This had been clarified in the amended statements [P 25, Line 89]

Editor Q5.

# Editor Q5. In your Data Availability statement, you have not specified where the minimal data set underlying the results described in your manuscript can be found. PLOS defines a study's minimal data set as the underlying data used to reach the conclusions drawn in the manuscript and any additional data required to replicate the reported study findings in their entirety. All PLOS journals require that the minimal data set be made fully available. For more information about our data policy, please see http://journals.plos.org/plosone/s/data-availability.

Reply

# In response to editor Q5: We had specified the minimal data set in the revised Data Availability statement [P 25, Line 92] and uploaded the data in the Supporting Information files.

Editor Q6.

# Editor Q6. Please include your tables as part of your main manuscript and remove the individual files. Please note that supplementary tables (should remain/ be uploaded) as separate "supporting information" files.

Reply

# In response to editor Q6: We had revised our manuscript to include our tables as part of our main manuscript [Table 1-5]

Reviewers' comments:

Reviewer's Responses to Questions

Comments to the Author

1. Is the manuscript technically sound, and do the data support the conclusions?

Reviewer #1: Yes

Reviewer #2: Yes

2. Has the statistical analysis been performed appropriately and rigorously? 

Reviewer #1: Yes

Reviewer #2: Yes

3. Have the authors made all data underlying the findings in their manuscript fully available?

Reviewer #1: Yes

Reviewer #2: Yes

4. Is the manuscript presented in an intelligible fashion and written in standard English?

Reviewer #1: Yes

Reviewer #2: Yes

5. Review Comments to the Author

Please use the space provided to explain your answers to the questions above. You may also include additional comments for the author, including concerns about dual publication, research ethics, or publication ethics. (Please upload your review as an attachment if it exceeds 20,000 characters) 

Reviewer #1: Not all lung adenocarcinomas with sensitive mutations can benefit from targeted therapy. How to distinguish effective and ineffective patients early in patients with targeted therapy has certain clinical significance. The author of the manuscript tried to answer this question through analyzing changes of tumor markers, which gave us some inspirations. There are three questions as follows. 

Comments to the Author

Reviwer-1 Q1.

Reviwer-1 Q1: Firstly, in manuscript, the authors evaluated the efficacy of targeted therapy by imaging-based response according to the RECIST standard. Why are the patients whose efficacy evaluation are SD divided into SD-20 and SD+30, instead of distinguishing by PR, SD, PD or DCR? It needs to be explained.

Reply

# In response to Reviewer-1 Q1: First, RECIST criteria have been used for evaluating the response of chemotherapy. However, if a targeted therapy for lung cancer responds to SD+29, clinicians may wonder whether this is an effective drug or not. Consequently, some studies used waterfall plots to reveal the response of targeted therapies. Furthermore, from a statistical point of view, it is difficult for PR, SD, or PD to correlate with the tumor marker's ascending or descending. On the contrary, the comparison of 2 * 2 (good/poor response VS. descending/ascending type) will be much more straightforward. Therefore, we combined categories of PR and SD-30 into the "good response group" due to reducing the tumor size. In contrast, the "poor response group" included cases with PD and SD+20 due to increased tumor size. These were clarified in the revised manuscript [method Page 10, Line 188-192; P11, Line 193-194]

Change in the text (See Page 10, Line 188-192; P11, Line 193-194) 

While analyzing the correlation between 4-TMpc after 14 days TKI targeted therapy and RECIST assessed response, and we combined categories of PR and SD-30 into the “good response group.” In contrast, the “poor response group” included cases with PD and SD+20. This classification's rationales were that: (1) It was the straight forward approach to classify responders vs. non-responders. (2) It was also difficult for PR, SD, or PD to correlate with the ascending or descending of the 4-TMpc from a statistical point of view.

Reviwer-1 Q2.

# Reviewer-1 Q2: Secondly, the references at the end of the manuscript seem to be inconsistent with the label in the article. For example, the reference 11 in the article seems to be 13 at the end, and it needs to be revised.

Reply

# In response to Reviewer-1 Q2: I am sorry for the apparent mistake and thank the reviewer for the careful review. I had corrected it!

Change in the text (See Page 22, Line 21-22) 

Zhang et al. indicated that the descending type of CEA within one month correlated with PR and SD of EGFR-TKI in patients with lung adenocarcinoma [13].

13. Zhang Y, Jin B, Shao M, Dong Y, Lou Y, et al. (2014) Monitoring of carcinoembryonic antigen levels is predictive of EGFR mutations and efficacy of EGFR-TKI in patients with lung adenocarcinoma. Tumor Biology 35: 4921-4928.

Reviwer-1 Q3

#Reviewer-1 Q3: Thirdly, cases in this study are all from 2012 to 2015, why are there no recent patients enrolled to increase the sample size for further statistical analysis?

Reply

# In response to Reviewer-1 Q3: We thanked the reviewer for this comment. We used cases within 2012 to 2015 because the enrolled cases and the follow-up time are sufficient to reflect the study results. [Also see the above “in response to editor Q2”]. Because TKI targeted therapy was still the current standard of care for these patients, we believed our study results could help early therapeutic decision making by identifying patients who may benefit from TKI targeted therapy [Discussion; P24, Line 54-61]

Change in the text (Page 24, Line 54-61) 

We did not extend our study after Feb 2015 because the enrolled cases and the follow-up time are sufficient to reflect the study results in which we did find high-level agreement [kappa= 0.762] by planned accrual [around 190 cases]. Furthermore, the investigated scenario (gefitinib, erlotinib, afatinib, crizotinib or ceritinib for lung adenocarcinoma patients) was still valid in Taiwan in 2020, although whether our finding was applicable for some new inhibitors (such as osimertinib or alectinib) deserved to be studied.

Reviewer #2: It is difficult to evaluate efficacy of treatment in patients with lung cancer in some cases. The present study　investigated the association between TKI treatment and tumor markers in advanced lung cancer. Overall, the study is well performed, however, there are potential limitations in this study. 

Reviwer-2 Q1.

# Reviewer-2 Q1: First, there was no mention of the reason why the authors chose those tumor markers. 

Reply

# In response to Reviewer-2 Q1: We thanked the reviewer for this comment. This had been clarified in the revised manuscript [Introduction Page 6, Line 93-96]..

Change in the text (See Page 6, Line 93-96) 

The selection of these tumor markers was based on reports [9-14] and our previous pilot study results (data not shown). There 4-TMs were also easy to assess in clinical practice and provided the most cost-effective coverage in patients with advanced lung adenocarcinoma.

9. Chiu C-H, Shih Y-N, Tsai C-M, Liou J-L, Chen Y-M, et al. (2007) Serum tumor markers as predictors for survival in advanced non-small cell lung cancer patients treated with gefitinib. Lung Cancer 57: 213-221.

10. Fritsche HA (1993) Serum tumor markers for patient monitoring: a case-oriented approach illustrated with carcinoembryonic antigen. Clin Chem 39: 2431-2434.

11. Ferrigno D, Buccheri G (1995) Clinical applications of serum markers for lung cancer. Respir Med 89: 587-597.

12. Grunnet M, Sorensen J (2012) Carcinoembryonic antigen (CEA) as tumor marker in lung cancer. Lung Cancer 76: 138-143.

13. Zhang Y, Jin B, Shao M, Dong Y, Lou Y, et al. (2014) Monitoring of carcinoembryonic antigen levels is predictive of EGFR mutations and efficacy of EGFR-TKI in patients with lung adenocarcinoma. Tumor Biology 35: 4921-4928.

14. Facchinetti F, Aldigeri R, Aloe R, Bortesi B, Ardizzoni A, et al. (2015) CEA serum level as early predictive marker of outcome during EGFR-TKI therapy in advanced NSCLC patients. Tumour Biol 36: 5943-5951.

Reviwer-2 Q2.

#Reviewer-2 Q2: Second, a method of obtaining pathological samples in patients with lung cancer was not described in this study. 

Reply

# In response to Reviewer-2 Q2: We thanked the reviewer to point this issue. This had been clarified in the revised manuscript [Results Page 12, Line 218-221].

Change in the text (See Page 12, Line 218-221) 

……184 patients and 191 patient-times were enrolled in this study, including 29 accepted the diagnostic procedure of computed tomography-guided core needle biopsy, 68 accepted transbronchial biopsy, 20 accepted ultrasound-guided biopsy, 27 accepted thoracentesis or pericardiocentesis, and 40 accepted operative procedures.

# Reviwer-2 Q3.

# Reviewer-2 Q3: Third, some patients received TKIs regardless of unknown mutation status. Is it ethically acceptable?

Reply

# In response to Reviewer-2 Q3: We thanked the reviewer for addressing this issue. We had clarified that this strategy is ethically acceptable in Taiwan. We included 28% of patients without mutation or with unknown mutation status. The rationale is that a portion of patients with EGFR wild type mutations and ALK-negative have shown a response to TKI targeted therapies [1,2]. Besides, our study was approved by the Institutional Review Board with written informed consent from all patients, and this approach was also reimbursed by Taiwan National Health insurance for selected patients.

1. John T, Liu G, Tsao MS (2009) Overview of molecular testing in non-small-cell lung cancer: mutational analysis, gene copy number, protein expression and other biomarkers of EGFR for the prediction of response to tyrosine kinase inhibitors. Oncogene 28 Suppl 1: S14-23.

2. Shaw AT, Ou SH, Bang YJ, Camidge DR, Solomon BJ, et al. (2014) Crizotinib in ROS1-rearranged non-small-cell lung cancer. N Engl J Med 371: 1963-1971.

6. PLOS authors have the option to publish the peer review history of their article (what does this mean?). If published, this will include your full peer review and any attached files.

Do you want your identity to be public for this peer review? For information about this choice, including consent withdrawal, please see our Privacy Policy.

Reviewer #1: Yes: Jinliang Wang

Reviewer #2: No

---

## [Decision Letter · Decision Letter 1]

2 Oct 2020

Early Serum Tumor Marker Levels After Fourteen Days of Tyrosine Kinase Inhibitor Targeted Therapy Predicts Outcomes in Patients with Advanced Lung Adenocarcinoma

PONE-D-20-12186R1

Dear Dr. Dr. Chen,

We’re pleased to inform you that your manuscript has been judged scientifically suitable for publication and will be formally accepted for publication once it meets all outstanding technical requirements.

Kind regards,

Christina L Addison, Ph.D.

Academic Editor

PLOS ONE

Additional Editor Comments (optional):

Reviewers' comments:

Reviewer's Responses to Questions

**Comments to the Author**

1. If the authors have adequately addressed your comments raised in a previous round of review and you feel that this manuscript is now acceptable for publication, you may indicate that here to bypass the “Comments to the Author” section, enter your conflict of interest statement in the “Confidential to Editor” section, and submit your "Accept" recommendation.

Reviewer #1: All comments have been addressed

Reviewer #2: All comments have been addressed

2. Is the manuscript technically sound, and do the data support the conclusions?

Reviewer #1: Yes

Reviewer #2: Yes

3. Has the statistical analysis been performed appropriately and rigorously? 

Reviewer #1: Yes

Reviewer #2: Yes

4. Have the authors made all data underlying the findings in their manuscript fully available?

Reviewer #1: Yes

Reviewer #2: Yes

5. Is the manuscript presented in an intelligible fashion and written in standard English?

Reviewer #1: Yes

Reviewer #2: Yes

6. Review Comments to the Author

Reviewer #1: Not all lung adenocarcinomas with sensitive mutations can benefit from targeted therapy. How to distinguish effective and ineffective patients early in patients with targeted therapy has certain clinical significance. The author of the manuscript tried to answer this question through analyzing changes of tumor markers, which supplied us a new method.

Reviewer #2: (No Response)

7. PLOS authors have the option to publish the peer review history of their article (what does this mean?). If published, this will include your full peer review and any attached files.

Reviewer #1: No

Reviewer #2: **Yes: **Fumihiro Yamaguchi

---

## [Editor Report · Acceptance letter]

13 Oct 2020

PONE-D-20-12186R1 

Early Serum Tumor Marker Levels After Fourteen Days of Tyrosine Kinase Inhibitor Targeted Therapy Predicts Outcomes in Patients with Advanced Lung Adenocarcinoma 

Dear Dr. Chen:

I'm pleased to inform you that your manuscript has been deemed suitable for publication in PLOS ONE. Congratulations! Your manuscript is now with our production department. 

Kind regards, 

on behalf of

Dr. R&R PLOS 

Staff Editor

PLOS ONE